# The Influence of Colloidal Properties of Carbon Black on Static and Dynamic Mechanical Properties of Natural Rubber

**DOI:** 10.3390/polym14061194

**Published:** 2022-03-16

**Authors:** William Amoako Kyei-Manu, Charles R. Herd, Mahatab Chowdhury, James J. C. Busfield, Lewis B. Tunnicliffe

**Affiliations:** 1School of Engineering and Materials Science, Queen Mary University of London, London E1 4NS, UK; w.a.kyei-manu@qmul.ac.uk; 2Birla Carbon, Marietta, GA 30062, USA; charles.herd@adityabirla.com (C.R.H.); mahatab.chowdhury@adityabirla.com (M.C.); lewis.tunnicliffe@adityabirla.com (L.B.T.)

**Keywords:** carbon black, elastomer, rubber, tensile, Mullins effect, Payne effect, dynamic strain, hysteresis, natural rubber

## Abstract

The influence of carbon black (CB) structure and surface area on key rubber properties such as monotonic stress-strain, cyclic stress–strain, and dynamic mechanical behaviors are investigated in this paper. Natural rubber compounds containing eight different CBs were examined at equivalent particulate volume fractions. The CBs varied in their surface area and structure properties according to a wide experimental design space, allowing robust correlations to the experimental data sets to be extracted. Carbon black structure plays a dominant role in defining the monotonic stress–strain properties (e.g., secant moduli) of the compounds. In line with the previous literature, this is primarily due to strain amplification and occluded rubber mechanisms. For cyclic stress–strain properties, which include the Mullins effect and cyclic softening, the observed mechanical hysteresis is strongly correlated with carbon black structure, which implies that hysteretic energy dissipation at medium to large strain values is isolated in the rubber matrix and arises due to matrix overstrain effects. Under small to medium dynamic strain conditions, classical strain dependence of viscoelastic moduli is observed (the Payne effect), the magnitude of which varies dramatically and systematically depending on the colloidal properties of the CB. At low strain amplitudes, both CB structure and surface area are positively correlated to the complex moduli. Beyond ~2% strain amplitude the effect of surface area vanishes, while structure plays an increasing and eventually dominant role in defining the complex modulus. This transition in colloidal correlations reflects the transition in stiffening mechanisms from flexing of rigid percolated particle networks at low strains to strain amplification at medium to high strains. By rescaling the dynamic mechanical data sets to peak dynamic stress and peak strain energy density, the influence of CB colloidal properties on compound hysteresis under strain, stress, and strain energy density control can be estimated. This has considerable significance for materials selection in rubber product development.

## 1. Introduction

The use of carbon black (CB) as a reinforcing agent for rubber allows for very precise tuning of rubber compound behavior via the appropriate selection of the colloidal properties and loading level of the CB in the compound. The incorporation of CB into rubber affects practically every aspect of rubber behavior. Properties of particular interest include the stress–strain and dynamic mechanical behavior of rubbers. These properties have major influences on final product performance, including the static and dynamic stiffness and deflection of the rubber under various loading conditions, material compliance and traction (in the case of rubber–surface contact, experienced for example in tires and dynamic/static seals), and the heat buildup and mechanical energy dissipation of the rubber product, which is of particular concern for tire fuel efficiency. These properties play a critical role in defining the fatigue and lifetime performance of rubber products, although fatigue performance is beyond the scope of this particular work.

A substantial body of historical experimental work has been performed in order to understand the role of CB in rubber reinforcement, particularly in regard to selection of CB and the effects of CB on the functional and material properties of rubber compounds [1,2,3,4,5,6,7,8,9]. However, large gaps in the understanding of the exact mechanisms of reinforcement persist. Specifically, the mechanisms by which rubber is stiffened by incorporation of CB and other particulates as well as the strain history (Mullins Effect) and dynamic strain amplitude dependence (Payne Effect) of CB-reinforced rubbers remain incompletely understood, despite being of broad industrial significance. The basic stiffening effect of CB has to various degrees been attributed to strain amplification/overstrain of the rubber matrix [10,11,12,13], particulate networking/flocculation [14,15,16], and modifications to the local dynamics of the rubber matrix [17,18], all of which are dependent on the strain, temperature, and strain rate conditioning of the rubber compound. The strain history, or Mullins Effect, which is typically measured at moderate to large strains, has been extensively investigated [19,20,21,22] and attributed to microstructural damage or reorganizations originating from strain amplification/overstrain of the rubber matrix [23], strain dependent rupture or damage of flocculated particle clusters (which may [17] or may not [24,25] be percolated via surface immobilized polymer fractions), and slippage or rupture of physically or chemisorbed rubber chain–CB bonds [3,26]. The Payne Effect is observed at low to moderate strains, typically by application of dynamic strain ramps, and has been attributed to a breakdown and reformation of a particle network within the rubber which is percolated via direct particle contacts, van der Waals interactions [14,27,28], or surface-immobilized rubber [17]. Conversely, it has also been proposed that labile connections between the rubber matrix and the particle surface are responsible for the Payne Effect [29,30]. A complete and fully accepted microstructural explanation of these various manifestations of particulate reinforcement of rubber is yet to be realised. It is important to note that any such explanation should be capable of describing all observed phenomenology of rubber reinforcement. To that end, it is important to comprehensively map out the influence of CB colloidal properties on these manifestations of rubber reinforcement.

The fundamental particulate unit of carbon black is the aggregate, which is formed by a fused assembly of broadly spherical para-crystalline primary particles with diameters ranging from ~200 nm to ~5 nm. The number and spatial arrangement of primary particles comprising the aggregate define its “structure” level. Particle size and structure level are parameters which can be independently controlled during production of carbon black in furnace reactors. Particle size can be measured directly using transmission electron microscopy [31] or inferred from bulk measurements of surface area using gas adsorption techniques [32]. Structure is typically measured by oil adsorption tests [33,34]. These parameters play a key role in defining the levels of reinforcement imparted to a rubber compound by carbon black. For example, primary particle size is the key parameter defining both the contact area between CB and rubber and the number of aggregates per unit volume of rubber, and therefore governs the average inter-aggregate distance and aggregate–aggregate “networking”. From simple geometrical considerations, the number density of primary CB particles per unit volume of rubber compound scales with the cube of the surface area. The aggregate structure is related to the volume of rubber occluded or screened from globally-applied strains by aggregate branches [30,35]. The effective volume fraction of solid in a rubber compound is therefore the sum of the CB volume and the volume of occluded rubber, with the latter directly related to CB structure level [35,36,37]. This has a direct impact on levels of strain amplification in the compound. Other important parameters of carbon black include surface chemistry/activity [3], porosity, thermal history [3,38], and the distributional nature of primary particle size and aggregate structure [37]. A detailed exploration of the effects of these parameters on rubber compound behavior is beyond the scope of this study.

In this work, we comprehensively examine the influence of CB surface area and structure on key properties of natural rubber compounds prepared with iso-loading (and volume fraction) of CB. A very wide colloidal space experimental design approach is taken, using furnace CBs varying only in their respective levels of structure and surface area. The wide colloidal space approach allows interpolation of our results and conclusions to cover the majority of industrial furnace CBs. From the resulting rubber compound data, new correlations and key insights can be drawn into the origins of CB reinforcement and design, and selection guides for CBs are consequently revisited.

## 2. Materials and Methods

### 2.1. Materials

Compounds of SMR CV60 natural rubber (NR) reinforced with eight different carbon black (CB) grades at 50 parts per hundred (phr) loading were prepared. An unreinforced NR counterpart was included in the tests for comparison. The CBs used in this study were selected to cover a broad range of surface area and structure, roughly correlating to a dual-factor central composite experimental design (as shown in Figure 1). Figure 1 shows several commonly used CB grades (N772, N660, N347, N330, N220, N115). These CBs are not evaluated in this work but are included in the figure to provide additional context. Table 1 shows the compound formulation used in this study. Table 2 provides the structure and surface area of the various CB grades. For the purposes of this paper, a naming convention is adopted which allows the reader to immediately identify the type of CB based on its CB structure and surface area; this is provided in Table 2 as well. The structure (measured by compressed oil absorption number, COAN) and surface area (measured by statistical thickness surface area, STSA) values of the CBs are listed as a superscript and a subscript, respectively. For example, N550, which has a structure value of 84 cc.100 g^−1^ and a surface area value of 37 m^2^·g^−1^, is referred to as ${\mathrm{CB}}_{37}^{84}$. The corresponding rubber compound produced using N550 is referred to by the same naming convention. Table 2 shows the interferometric microscope (IFM) dispersion index (DI) values of the final compounds measured according to ASTM D2663 (method D) [39]. Transmission electron microscopy (TEM) images of the tested carbon black samples are provided in the Appendix A.

Compounds were prepared by Birla Carbon (Marietta, GA, USA) using a 1.6 L capacity Banbury mixer. Vulcanized sheets measuring 11 mm × 11 mm × ~2 mm were prepared via compression molding at 150 °C for a time of T_90_ + 5 min where T_90_ (time at 150 °C required for the specimen to reach 90% maximum torque) was measured using a moving die rheometer (MDR) from Alpha Technologies located in Hudson, OH, USA. The mixing procedure used to prepare the compounds is provided in the Appendix A.

### 2.2. Shore A Hardness, Tensile to Break, and Cyclic Tensile Tests

Shore A hardness measurements were performed according to ASTM D2240 [40].

Dumbbells for uniaxial tensile testing were stamped from sheets of compound using a hydraulic die press. The dumbbells had approximate gauge length, width and thickness dimensions similar to ASTM D412 die C.

For uniaxial tensile testing to break, five dumbbell specimens were pulled until failure using a five station United tensile tester with a 1 kN load cell. Strain was defined using traveling contact extensometers; the extension rate was 500 mm/min (~strain rate of 0.19/s), following ASTM D412 [41].

For cyclic tensile tests, five dumbbells were extended at 500 mm/min to an initial target strain of 20% then retracted to 0% strain. The first cycle at this specified strain was followed by two cycles to the same target strain of 20%. The same specimen was then extended to a higher target strain of 50% for three cycles. This sequence was continued sequentially to higher target strains of 100%, 200%, and 300%.

All tensile and hardness testing was performed at 21 °C ± 1 °C, 55% relative humidity, and atmospheric pressure conditions.

### 2.3. Dynamic Strain Sweep Characterization

Dynamic strain sweeps between 0.1% and 62.5% single strain amplitude were performed at 10 Hz with zero mean strain using an ARES G2 torsional rheometer from TA Instruments located in New Castle, DE, USA at 60 °C. Specimen geometries were cylinders measuring 8 mm in diameter and with ~2 mm thickness, which were stamped from sheets of vulcanized compound and bonded to the rheometer parallel plate geometry using Loctite 480 adhesive by Henkel, Hemel Hempstead, UK. A slight compressive normal force of 100 g was applied to the cylinders during the test procedure. We note that this particular specimen geometry has a non-uniform strain field, varying with the radius of the cylinder during deformation; the reported strain amplitude values are for the extremity of the cylinder radius. The strain sweep test was performed by pre-conditioning the specimen six times at the specified dynamic strain amplitude before collecting torque–time data. This process was repeated from low to high strain amplitudes.

## 3. Results and Discussion

### 3.1. Compound Dispersion Index

Compound dispersion indices as determined by IFM are presented in Table 2. All compounds except ${\mathrm{CB}}_{96}^{55}$ and ${\mathrm{CB}}_{161}^{62}$ had DI values > 98, indicating nearly full macro-incorporation of the CB into the rubber. The compounds ${\mathrm{CB}}_{96}^{55}$ and ${\mathrm{CB}}_{161}^{62}$ contain the lowest structure CBs in the colloidal experimental design and have comparatively high surface area values. At a fixed surface area, lower structure CBs are notoriously more difficult to disperse in rubber due to (i) the higher number of attractive contacts between aggregates in pelletized CB prior to mixing on a unit volume basis and (ii) the resulting lower mix viscosities versus medium-high structure CBs [42]. Despite the somewhat lower DI values for ${\mathrm{CB}}_{96}^{55}$ and ${\mathrm{CB}}_{161}^{62}$, the compound dispersions are reasonable and well in line with dispersion indices observed for commercially prepared rubber compounds.

### 3.2. Tensile Stress Strain and Shore A Hardness Measurements

Figure 2 shows the five stress–strain to failure data sets for each rubber compound. From a visual examination of the data, it is clear that despite the CBs being at equivalent loading/volume fraction in the compounds, the colloidal differences between the CBs impart major differences to the observed stress–strain behavior. CBs with higher structure impart higher secant moduli values in comparison to CBs with lower structure. Table 3 summarizes the modulus at 300% (reported as stress value at 300% elongation), percent elongation at break, tensile strength, and Shore A hardness values of the CB-reinforced and gum NR specimens.

For a quantitative analysis, multiple linear regression analyses of tensile properties to CB colloidal properties were conducted (the NR gum compound data were not included in the regressions). Multiple regressions were performed via error minimization in Origin 2019 per Equation (1), where Y is the dependent property being analyzed, C is an intercept constant, $\beta_{St}$ is the coefficient of the structure of CB and $\beta_{SA}$ is the coefficient of the surface area of CB, and ϵ is the error term. The full regression results (intercept, coefficients, *p* values, and regression R^2^) are provided in Table 4. Note that coefficients with *p* values > 0.05 are highlighted in bold and are not statistically correlated to the observed parameter.

From the multiple regression results, it can be seen that the tensile modulus (here, the example is 300% modulus) is strongly correlated with the structure of CB. The structure coefficient is positive, indicating that increasing the structure of the CB increases the resulting tensile modulus. By contrast, the CB surface area is less strongly correlated (higher *p* value) and has only a slight negative correlation coefficient with tensile modulus, meaning that increasing the surface area actually very slightly decreases the resulting tensile modulus. The fact that CB structure is the key parameter controlling stress–strain moduli is quite well known and is typically attributed to strain amplification/overstraining of the rubber matrix through occlusion/screening of a certain volume of rubber by the CB aggregate structure [7,8,37]. The effect by which increased surface area slightly reduces modulus values has been tentatively attributed to increasing deactivation of the cure system, for example, by the adsorption of accelerators on the surface of CB and consequent net reduction in compound crosslink density [43].

In addition, tensile failure parameters are correlated to varying extents with CB colloidal properties. Tensile strength increases with increasing CB surface area, likely due to an increase in compound critical tear energy [44]. Structure and surface area have opposing correlations with elongation at break. Higher structure CBs (which produce higher modulus compounds) reach their work at fracture at lower strains, while increasing CB surface area increases tensile strength and therefore increases elongation at break.

Shore A hardness is only correlated with structure, reflecting the predominant trend for the tensile moduli.
(1)$$Y=C+\beta_{St}\left(COAN\right)+\beta_{SA}\left(STSA\right)+\epsilon$$


### 3.3. Cyclic Tensile Tests

Figure 3 shows cyclic tensile data for the various rubber compounds. All compounds display evidence of the classical Mullins strain history effect to varying extents. Following initial strain cycles, the compounds display a pronounced softening effect upon subsequent cycling. When the specimen is stretched to higher peak strains than the previous cycles, the initial (virgin) stress–strain curve is recovered. Upon retraction, there are set effects (residual extension remaining) as well as evident mechanical hysteresis between the loading and unloading curves. As observed for the uniaxial tensile tests to break, CBs with higher structure impart a higher initial modulus to the compounds. Compounds containing higher structure CBs show a larger relative drop in stress–strain properties following the initial strain cycles than those containing lower structure CBs. From an initial visual examination, the role of CB particle size/surface area is not obvious.

In order to quantify the magnitude of these softening effects as a function of strain and cycle number, the mechanical hysteresis apparent between loading and unloading of the compounds was calculated as the difference between the respective integrals. Figure 4A–C show the mechanical hysteresis as a function of the first, second, and third strain cycles for each peak tensile strain. As expected, the magnitude of hysteresis increases with increasing tensile strain. The unfilled rubber shows the lowest levels of hysteresis, though nevertheless appreciable. The CB reinforced compounds vary substantially in the levels of hysteresis observed, with the ranking appearing to be predominantly dependent on CB structure. The magnitude of hysteresis observed at a given peak strain reduces upon sequential strain cycling, while the ranking of compounds remains consistent. The influence of the colloidal properties of CB on the magnitude and strain dependence of the mechanical hysteresis was evaluated by performing multiple linear regression analyses of hysteresis values at each peak strain level compared to the structure and surface area of the carbon blacks. The full results of these regression analyses are provided in the Appendix A. Figure 4D shows the multiple regression coefficients of both the structure and surface area of CB at each peak strain. Data points are solid where their regression *p* values are <0.05 and dashed when *p* values are >0.05. CB surface area is only correlated with hysteresis at the lowest strain levels, whereas structure is strongly correlated across the entire strain history range. This finding has a number of implications:At a practical level, the degree of mechanical hysteresis (and therefore softening) of a rubber compound at a fixed strain level scales with the virgin modulus of the compound at that strain. All other parameters being equal (such as CB volume fraction, polymer type, and crosslink density), this modulus is determined by the structure of the CB in the formulation [45].At a microstructural level, the strong correlation between hysteresis and CB structure provides several hints as to the origin of the Mullins-type hysteresis and softening. It suggests that the hysteretic energy dissipation at these large strains is isolated in the rubber matrix and arises due to strain amplification/matrix overstrain, as opposed to hysteretic polymer–particle surface slippage and/or hysteretic breakup of flocculated aggregate clusters, which have been proposed in the literature. Note that strain amplification as described by hydrodynamic-type equations is independent of CB particle size/surface area, which is consistent with our observations [10,11,12]. In these experiments, specimens have been cycled to specified strain levels. Harwood, Mullins, and Payne [23] conducted highly relevant experiments where specimens were cycled to specified stress levels. Under these conditions, the resulting mechanical hysteresis values were found to be identical for a wide range of CB reinforced and gum NR compounds. These findings are consistent with our results in the sense that they can both be explained if we assume that energy dissipation occurring at these large strains is isolated predominantly in the overstrained rubber matrix.

### 3.4. Dynamic Strain Sweeps

Figure 5 shows example stress–strain-time raw data at selected strain amplitudes collected for the ${\mathrm{CB}}_{111}^{107}$ specimen. As can be seen, the stress–strain response of the material is elliptical and the stress–time response shows sinusoidal behavior, even up to relatively large strain amplitudes. This is both an unusual aspect of rubber rheology (as most complex materials show non-sinusoidal distortions in their dynamic mechanical behavior at moderate-large strains [46]) and a highly advantageous aspect, as it allows application of linear viscoelasticity for analysis of commercially relevant materials at commercially relevant deformation amplitudes [15,28,47,48].

Key linear viscoelastic parameters and interrelationships are provided by Equations (2)–(6), assuming a strain-controlled experiment with application of a sinusoidal shear strain amplitude $\gamma_{0}$ at fixed frequency ω, eliciting a time dependent stress response, $\sigma \left(t\right)$, which can be decomposed into elastic and loss moduli ($G^{\prime }$ and G″, respectively). The magnitude of the complex modulus $\left|G^{*}\right|$, loss tangent tanδ, and loss compliance J″ values can then be defined in terms of the elastic and loss moduli as follows:(2)$$\gamma \left(t\right)\;=\gamma_{0}\sin\left(\omega t\right)$$
(3)$$\sigma \left(t\right)\;=\gamma_{0}\left[G^{\prime }\left(\omega \right)\sin\omega t+G\prime \prime \left(\omega \right)\cos\omega t\right]$$
(4)$$\left|G^{*}\right|=\sqrt{G\prime^{2}+G{\prime \prime }^{2}}$$
(5)$$\tan\delta =\frac{G\prime \prime }{G\prime }$$
(6)$$J^{\prime \prime }=\frac{G\prime \prime }{G\prime^{2}+G{\prime \prime }^{2}}=\frac{G\prime \prime }{{\left|G^{*}\right|}^{2}}$$

Plots of $\left|G^{*}\right|$ versus strain amplitude for each compound in this study are presented in Figure 6A. The strain dependence of the viscoelastic parameters of CB reinforced rubbers versus unreinforced rubber is widely known as the Payne effect, and is ascribed to dynamic breakdown and reformation of a particle network within the rubber compound. The detailed physics and micro-mechanics of such networks of particles remain rather poorly understood. The complex moduli data in Figure 6A exhibit several interesting features. There are large variations in the magnitude and ranking of $\left|G^{*}\right|$ at the smallest strain amplitudes, dependent on the colloidal properties of the CB. However, the ranking of $\left|G^{*}\right|$ magnitude changes substantially as the strain amplitude is increased, and data set “crossover” effects are observed.

The influence of the colloidal properties of CB on the magnitude and strain dependence of the complex moduli were evaluated by performing multiple linear regression analyses of $\left|G^{*}\right|$ values at each strain amplitude as compared to the structure and surface area of the carbon blacks. The full results of these regression analyses are provided in the Appendix A. Figure 6B shows the multiple regression coefficients of both the structure and surface area of CB at each strain amplitude. At low strain amplitudes (0.1–2%), both the structure and surface area of CB are positively correlated with $\left|G^{*}\right|$ to similar extents. The magnitudes of the coefficients decrease with increasing strain amplitude, and structure starts to dominate over surface area at higher strain amplitudes on a relative basis. With increasing strain amplitude, the coefficient of surface area to $\left|G^{*}\right|$ reduces to zero, while structure plays an increasing and eventually dominating role in defining $\left|G^{*}\right|$. In Figure 6B, coefficient data points are solid where their regression *p* values are <0.05 and dashed when *p* values are >0.05; *p* values for both structure and surface area are <0.05 at strain amplitudes <2%, indicating strong statistical correlation of both colloidal properties to $\left|G^{*}\right|$. After ~2% strain amplitude, the *p* value of the surface area steadily increases with increasing strain amplitude until it exceeds 0.05 at higher strains, indicating no statistically significant correlation with $\left|G^{*}\right|$.

Therefore, with increasing strain amplitude there is a clear transition of the colloidal properties of CB controlling $\left|G^{*}\right|$, and by extension, other viscoelastic parameters. At the microstructural level this transition can be interpreted as the result of the strain-dependent breakdown of rigid particle networks, with the degree of particle networking being governed by the number of aggregates per unit volume of rubber, which at fixed volume fraction is roughly the cube power of the surface area. This continues until at higher strains the primary stiffening mechanism becomes that of strain amplification alone, as defined by solid and occluded rubber fractions, CB loading, and structure level. This observed transition is consistent with the earlier observation that CB structure is strongly correlated with the tensile stress–strain moduli, which are measured at larger strain ranges than in the dynamic experiments.

At a practical level, this separability of surface area and structure effects on $\left|G^{*}\right|$ depending on the applied strain level allows for precise engineering of rubber compounds with strain-dependent viscoelastic moduli via appropriate selection of CB surface area and structure.

In service, rubber components are deformed under conditions of either strain, stress, strain energy control, or more commonly in complex combinations of several of these conditions. This has enormous practical consequences for materials selection and component design, particularly as relates to energy dissipation and fatigue life performance [49,50]. Under constant cyclic strain amplitude $\gamma_{0}$, stress amplitude $\sigma_{0}$, and strain energy density amplitude $W_{0}$, the energy dissipation density, $W_{d}$, can be derived from linear viscoelastic parameters (see Appendix A), yielding Equations (7)–(9), respectively:(7)$$W_{d}\left(\gamma_{0}\right)=\pi \gamma_{0}^{2}G\prime \prime$$
(8)$$W_{d}\left(\sigma_{0}\right)=\pi \sigma_{0}^{2}J\prime \prime$$
(9)$$W_{d}\left(W_{0}\right)\propto \tan\delta$$

The hysteresis performance of the rubber compounds under various deformation controls can therefore be predicted by comparison of the appropriate viscoelastic parameter at the equivalent deformation control condition (stress, strain, strain energy density).

For strain control, per Equation (7), compound hysteresis scales according to the loss modulus. Figure 7A shows the loss moduli derived from the raw data of the strain sweep experiments via Equations (2) and (3). The loss moduli data exhibit a pronounced strain dependence, with a peak in magnitude at around 2% strain amplitude. The magnitude of the G″ values vary substantially depending on the exact colloidal properties of the CB. Figure 7B shows the regression coefficients of surface area and structure for G″ values at each strain amplitude from multiple linear regression analyses; the full results of the regression analyses are provided in the Appendix A. As can be seen, both the surface area and structure of CB are positively correlated with G″, with surface area having the larger contribution at lower strains and structure having the larger contribution at high strains, similar to the observed correlations for the complex moduli. Therefore, in order to minimize compound hysteresis in strain control, CB with low surface area and low structure should be selected, with the resulting compound dynamic stiffness being reduced. A particularly interesting data set in Figure 7A is ${\mathrm{CB}}_{161}^{62}$, which contains a CB having very high surface area and very low structure. This compound has the highest G″ value at low to medium strains, owing to the significant stiffening effect of the surface area contribution, and a mid-ranked G″ value at high strains, where the more limited contribution of its structure to G″ dominates.

Characterizations of rubber material viscoelasticity are typically performed in strain control; however. it is possible to re-scale data collected in strain control to peak stress and peak strain energy density values collected during testing in order to obtain insight into the potential compound performance in other modes of deformation control. There are two key differences between this re-scaling approach and the direct collection of stress and strain energy density-controlled experimental data: (i) strain rate is controlled as opposed to stress rate or energy rate; and (ii) deformation history (the sequence of application of deformation cycles during the experiment) is strain controlled as opposed to stress or energy controlled. These limitations should be borne in mind for the proceeding analysis; the ideal situation is full experimental characterization of rubber compounds under each mode of deformation control.

Figure 8A shows the peak oscillatory stress values measured at each strain amplitude for each compound. Figure 8B shows the corresponding strain energy densities at each strain amplitude for each compound, which are calculated by integration of the data sets in Figure 8A. These data can be used to re-scale the relevant viscoelastic parameters in an attempt to predict hysteresis performance for strain energy control and stress control from data collected under strain control.

For the case of strain energy density control, the loss tangent is predictive of hysteresis (Equation (9)). Figure 9A shows the loss tangent as a function of strain amplitude. The classical peak in loss tangent with increasing strain amplitude is observed for each compound. Figure 9B shows the loss tangent re-plotted as a function of peak strain energy density, and Figure 9C shows the coefficients for CB surface area and structure to tanδ as determined by multiple linear regressions carried out on data sets interpolated from Figure 9B. The full results of the regression analyses and examples of the interpolated tanδ–stress data sets are provided in the Appendix A. As can be seen from Figure 9C, only the surface area of CB is statistically correlated with tanδ over the majority of the strain energy density range. Therefore, in order to minimize hysteresis in strain energy density control, the CB surface area should be minimized.

For the case of stress control, loss compliance is predictive of hysteresis (Equation (8). Figure 10A shows loss compliance as a function of strain amplitude. Figure 10B shows loss compliance re-plotted as a function of peak stress amplitude. The loss compliance of the various compounds increases with increasing strain amplitude as the compound softens, and shows complex and stress-dependent material rankings over the range of peak stress values observed. Figure 10C shows the coefficients for CB surface area and structure as determined by multiple linear regressions carried out on data sets interpolated from Figure 10B. The full results of the regression analyses and examples of the interpolated J″-stress data sets are provided in the Appendix A. The regression results show a complex relationship between CB colloidal properties and J″.

At low stress levels (<0.1 MPa), both surface area and structure are negatively correlated with J″, implying that an increase in CB surface area and structure reduces J″, and therefore hysteresis as well. In this stress region, stiffer compounds experience less deflection, and the predicted hysteresis is therefore minimized. At larger stresses, the correlations diverge. Structure remains negatively correlated with J″ up to high strains, while surface area reverses sign and becomes positively correlated with J″. This complex dependence of J″ on CB colloidal properties can be rationalized by considering the definition of loss compliance (see Equations (4) and (6)). In order to reduce compound hysteresis, J″ must be minimized. Therefore, G″ must be minimized and $\left|G^{*}\right|$ maximized, which is a conflicting requirement. In the low stress region, maximizing $\left|G^{*}\right|$ requires selection of CB with high structure and high surface area (Figure 6B). Although this would increase G″, the $\left|G^{*}\right|$ term in Equation (6) is squared, and thus dominates J″ values in this stress range. At higher stresses, the contribution of surface area to $\left|G^{*}\right|$ is diminished (see Figure 6B), while for G″ it remains appreciable (Figure 7B), leading to the change in sign of the surface area coefficient. In this medium to high stress region, maximizing CB structure in order to increase stiffness and minimize compound deflection while minimizing surface area to suppress G″ yields lower J″, and therefore, hysteresis.

This nuanced result has implications for material selection tradeoffs when designing rubber components. Should the component be operated entirely in stress control, knowledge of the levels/distributions of applied stresses in the component is critical in order to enable optimum selection of CB properties. Should the component operate in a mix of stress, strain, and strain energy control, the contradictory requirements for optimizing hysteresis performance should be taken into account; finite element simulations of in-service component deformations can guide this process.

## 4. Conclusions

The effects of CB structure and surface area on the static and dynamic mechanical properties of rubber were studied; CB structure increases the stress–strain moduli due to overstraining effects in the rubber matrix, while increased CB surface area slightly reduces the observed moduli. The latter effect is proposed to be due to adsorption of accelerators on the surface of CB, resulting in a net reduction in matrix crosslink density. Mechanical hysteresis from cyclic tensile testing to fixed peak strain is predominantly controlled by CB structure. At a fixed strain level, the mechanical hysteresis, and therefore the softening of the rubber compound, scales with the compound’s virgin modulus at that strain. This suggests that the hysteretic energy dissipation at these large strains occurs in the rubber matrix and arises due to matrix overstrain effects rather than through rupture of flocculated particle clusters or interfacial polymer slippage. This is broadly in line with the conclusions drawn from cyclic experiments conducted to specified stress levels by Harwood, Mullins, and Payne. Further cyclic tensile testing of the compounds in the current study for comparison to controlled stress and controlled strain energy density levels would be highly informative in this regard. Dynamic mechanical properties show classical strain amplitude dependence (the Payne effect), which varies in magnitude according to the colloidal properties of CB. At low to medium strain amplitudes (0.1–2%), both the structure and the surface area of CB are positively correlated to $\left|G^{*}\right|$ to similar extents. Beyond 2%, however, the effect of surface area is reduced and structure plays an increasing and eventually dominant role in defining $\left|G^{*}\right|$. This transition in colloidal correlations is further evidence that a transition in stiffening mechanisms occurs with increasing strain amplitude. At low strains, the viscoelastic moduli are controlled by the flexing of percolated particle–particle networks, the formation of which is promoted by higher CB surface area. At higher strains, stiffening appears to be described best by matrix overstraining/strain amplification effects, which at iso-volume fractions are controlled by CB structure, and which map to correlations observed in the monotonic tensile stress–strain data. Replotting of the dynamic data versus peak stress and energy density allows an approximate assessment of the hysteresis response of the rubber compounds under various modes of deformation control. These results are summarized in Table 5, which presents proposed CB selection criteria for both dynamic and static conditions under iso-strain, iso-stress, and iso-strain energy density control. The table assumes a desire to minimize mechanical hysteresis; in practice, mechanical hysteresis is typically not the sole performance parameter guiding material selection. The need to achieve a specified compound tensile modulus (or hardness), tensile and tear strength, abrasion resistance, or friction performance level introduces material selection conflicts which run contrary to the trends shown in Table 5. Further forthcoming work on these materials will highlight these tradeoffs for this set of compounds, for example, in terms of their respective failure properties.

## Figures and Tables

**Figure 1 polymers-14-01194-f001:**
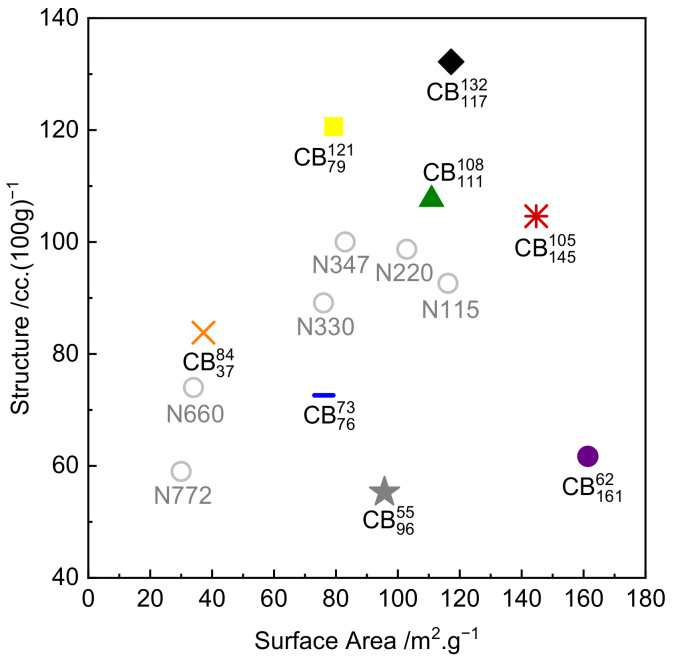
Colloidal plot of tested carbon blacks, with commonly used carbon black grades for reference (open circles).

**Figure 2 polymers-14-01194-f002:**
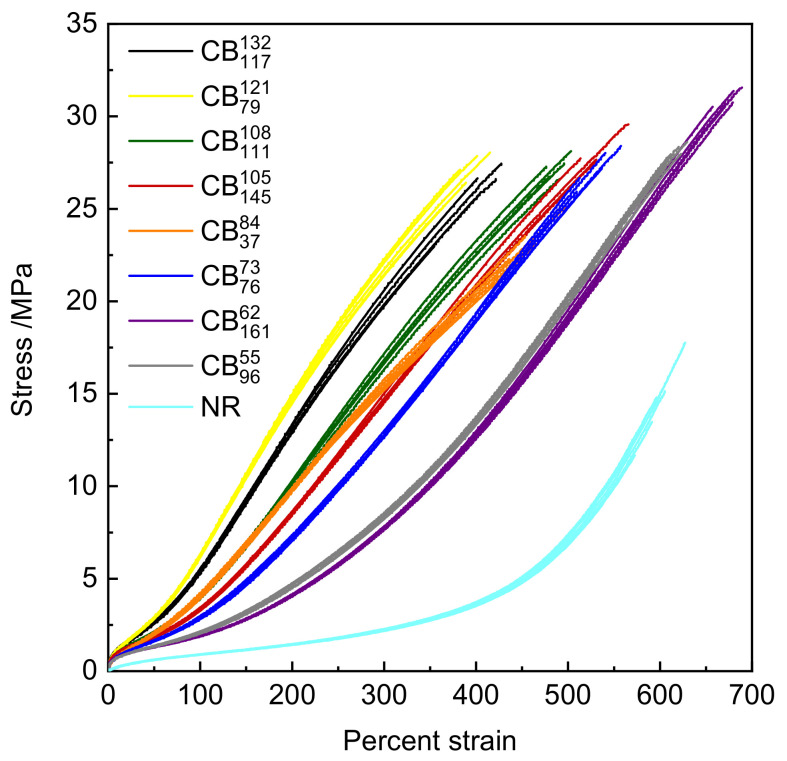
Stress–strain to failure data for the various rubber compounds.

**Figure 3 polymers-14-01194-f003:**
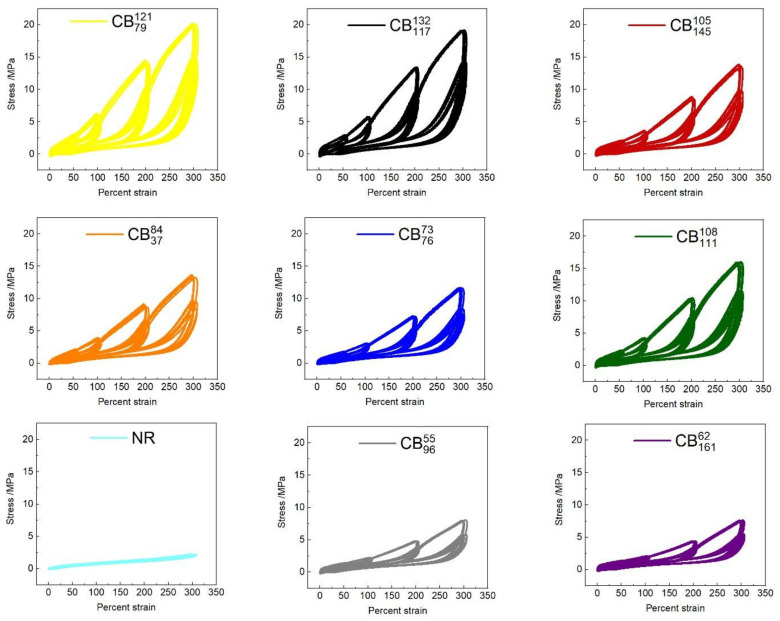
Stress–strain data, showing cyclic tensile test effects of the various rubber compounds.

**Figure 4 polymers-14-01194-f004:**
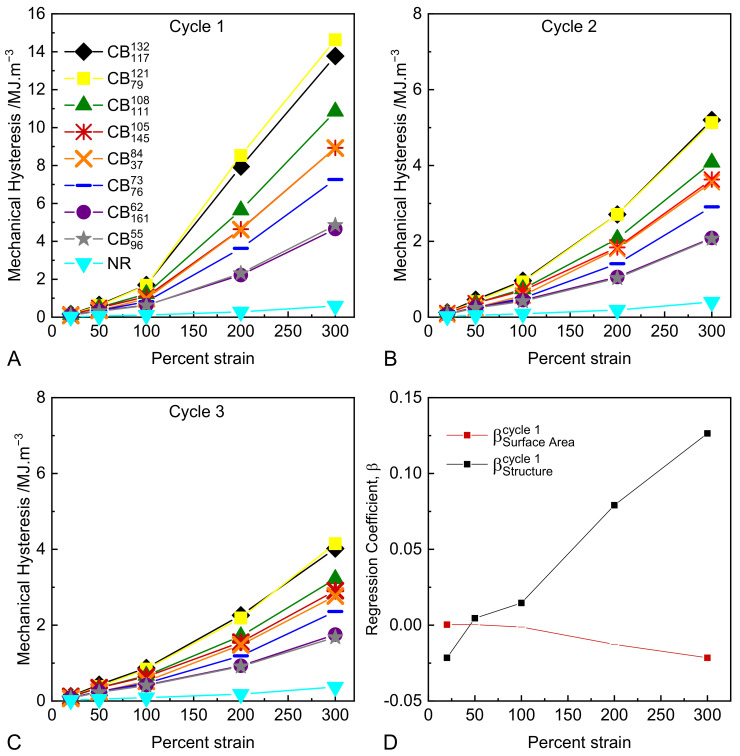
(**A**–**C**) Mechanical hysteresis after the first, second and third, strain cycles respectively (note the difference in ordinate scale between the first, second, and third cycle data); (**D**) regression coefficients of structure and surface area compared to mechanical hysteresis as measured on the first strain cycle (data points with *p* values > 0.05 are excluded).

**Figure 5 polymers-14-01194-f005:**
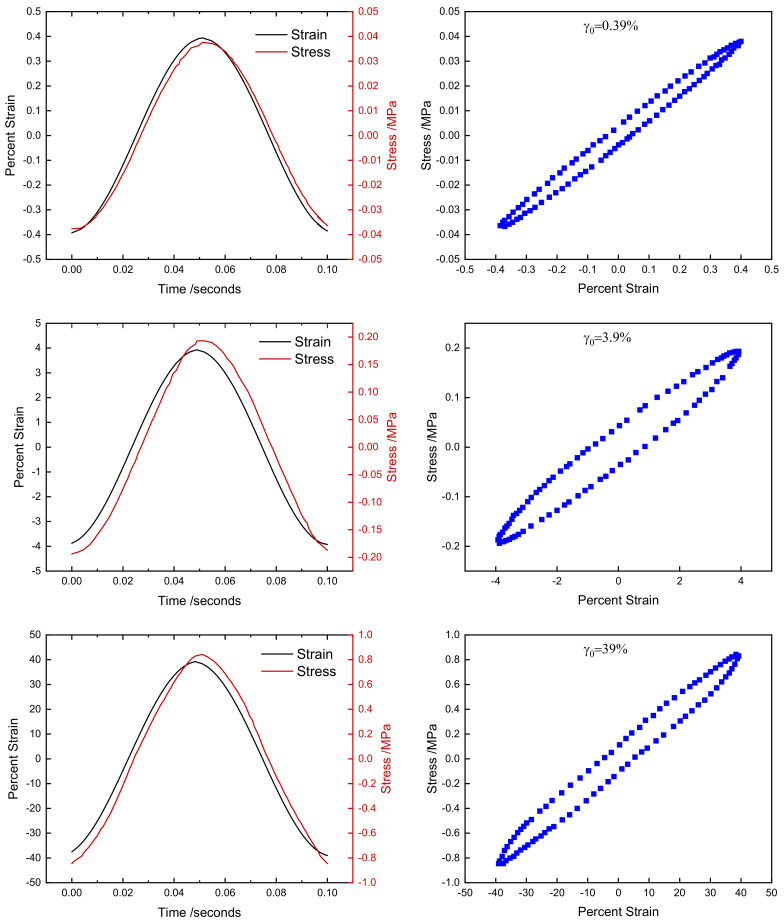
Stress–strain–time data for compound ${\mathrm{CB}}_{111}^{107}$ at selected strain amplitudes.

**Figure 6 polymers-14-01194-f006:**
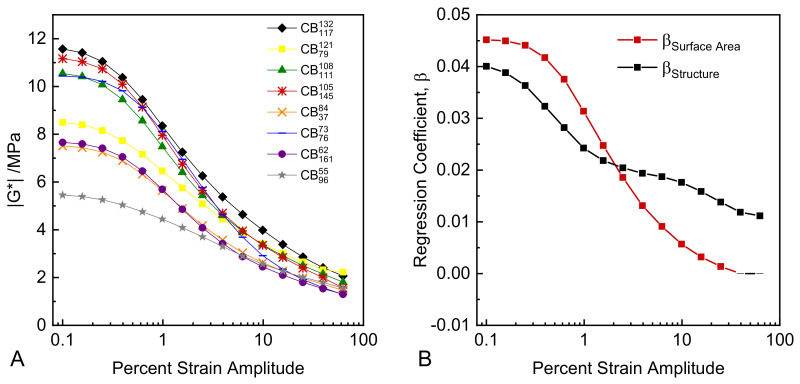
(**A**) $\left|G^{*}\right|$ versus strain amplitude; (**B**) multiple regression coefficients for structure and surface area versus strain amplitude (points with *p* values > 0.05 are shown as dashes).

**Figure 7 polymers-14-01194-f007:**
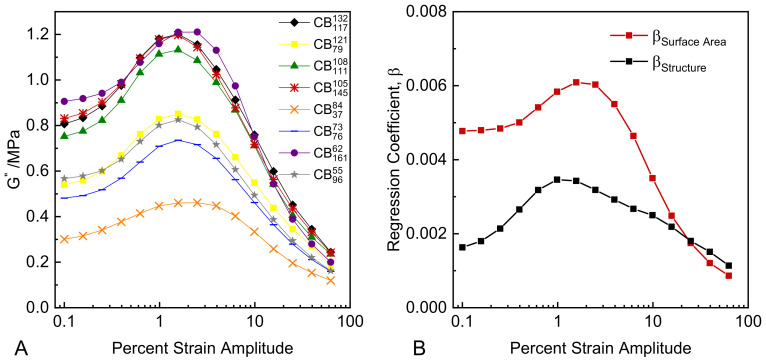
(**A**) G″ plotted versus strain amplitude; (**B**) multiple regression coefficients for structure and surface area plotted versus strain amplitude.

**Figure 8 polymers-14-01194-f008:**
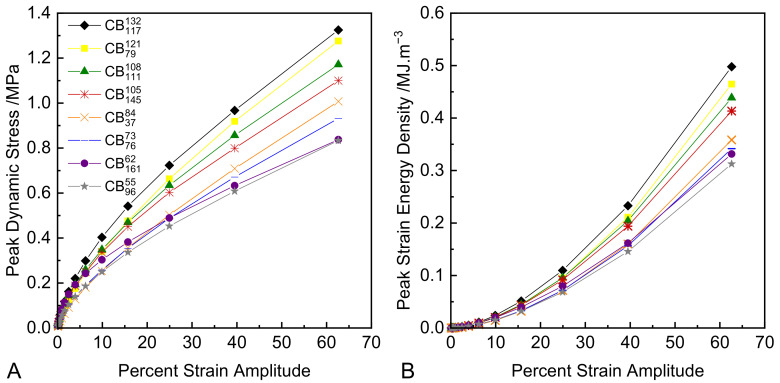
(**A**) Peak dynamic stress plotted versus strain amplitude; (**B**) peak dynamic strain energy density plotted versus strain amplitude.

**Figure 9 polymers-14-01194-f009:**
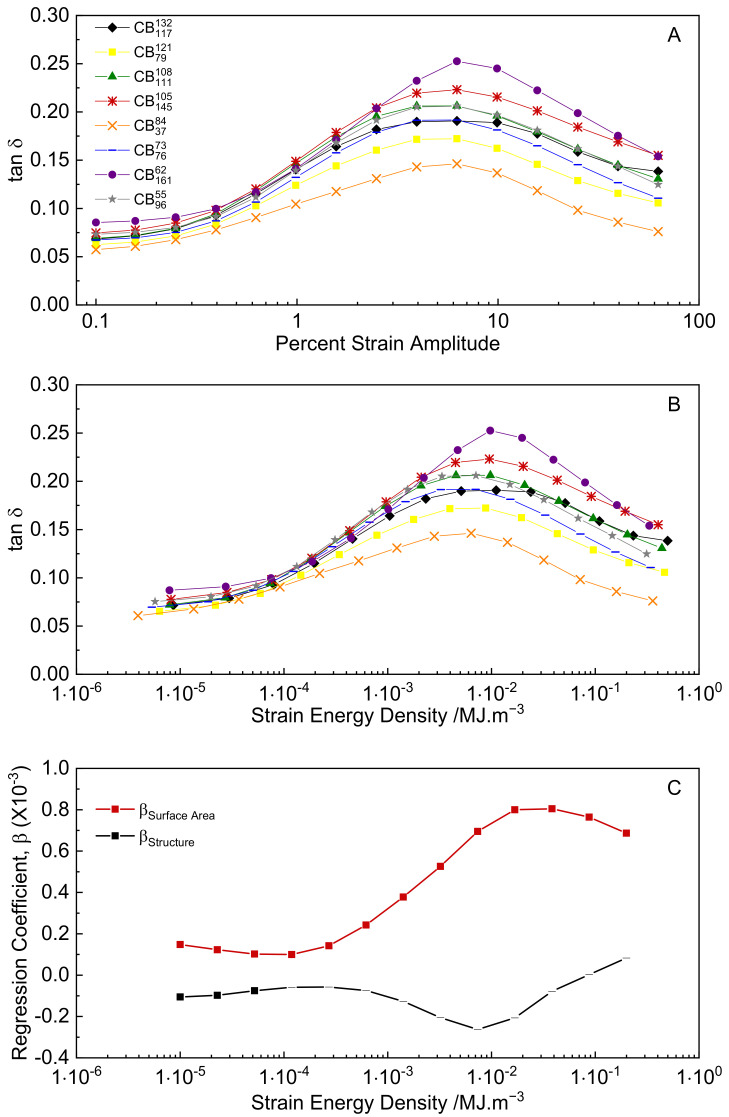
(**A**) tanδ versus strain amplitude; (**B**) tanδ versus peak dynamic strain energy density; (**C**) multiple regression coefficients for structure and surface area versus peak dynamic strain energy density (points with *p* values > 0.05 are shown as dashes).

**Figure 10 polymers-14-01194-f010:**
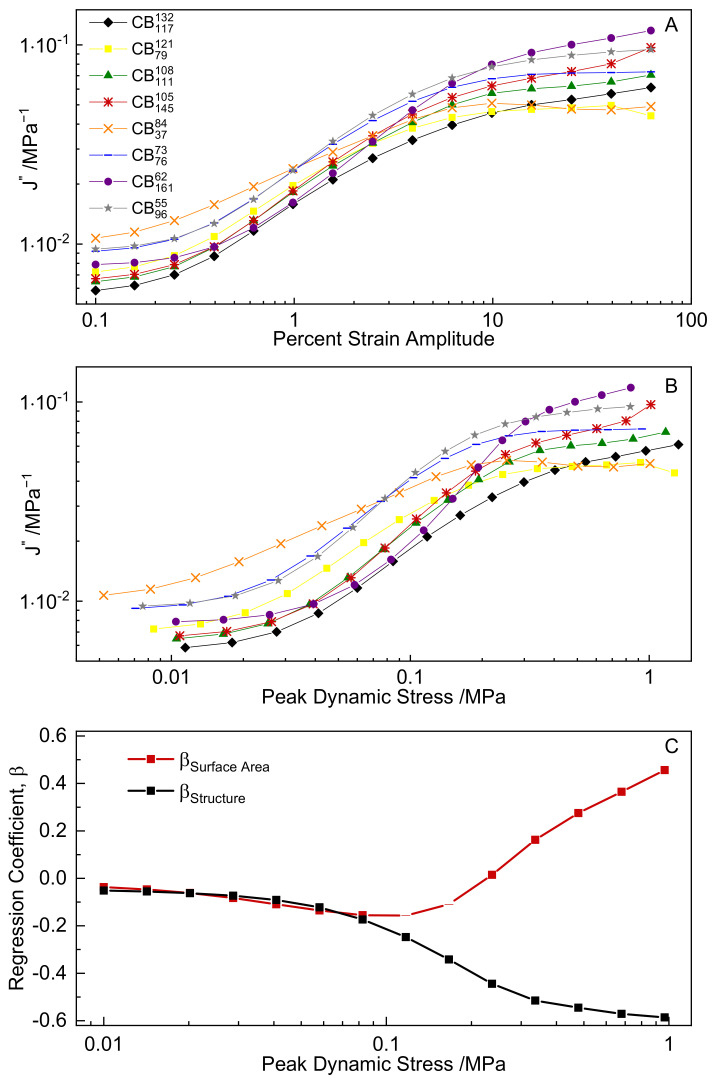
(**A**) J″ plotted versus strain amplitude; (**B**) J″ plotted versus peak dynamic stress; (**C**) multiple regression coefficients for structure and surface area plotted versus peak dynamic stress (points with *p* values > 0.05 are shown as dashes).

**Table 1 polymers-14-01194-t001:** Compound formulation.

Component	Loading/Parts per Hundred Rubber (phr)	Manufacturer of Component
NR—SMR CV-60	100	Herman Weber & Co.
Carbon Black	50	Birla Carbon
Zinc Oxide	5	Akrochem
Stearic Acid	3	PMC Biogenix
Anti-ozonant/Antioxidant	3	Americas International
Micro-wax	2	Strahl & Pitsch
Sulphur	2.5	R.E. Carroll
TBBS *-75	0.8	Akrochem

* *N*-Tertiarybutyl-2-benzothiazole sulfonamide, 75% assay.

**Table 2 polymers-14-01194-t002:** CB structure and surface area; CB naming convention adopted in this paper; compounds and compound dispersion index.

Carbon Black/ Compound Code	Structure (COAN)/ cc.(100 g)^−1^	Surface Area (STSA)/ m^2^·g^−1^	Carbon Black Commercial Name	Corresponding Compound Dispersion Index
Unfilled NR	NA	NA	NA	NA
${\mathrm{CB}}_{117}^{132}$	132	117	BC2005	99.3
${\mathrm{CB}}_{145}^{105}$	105	145	BC2115	98.8
${\mathrm{CB}}_{79}^{121}$	121	79	BC2013	98.8
${\mathrm{CB}}_{111}^{108}$	108	111	N234	99.4
${\mathrm{CB}}_{76}^{73}$	73	76	N326	98.0
${\mathrm{CB}}_{96}^{55}$	55	96	Raven 1200	90.2
${\mathrm{CB}}_{161}^{62}$	62	161	Raven 2000	81.5
${\mathrm{CB}}_{37}^{84}$	84	37	N550	98.7

**Table 3 polymers-14-01194-t003:** Summary of the hardness and tensile properties of the various rubber compounds.

Compound Code	300% Modulus /MPa	Percent Elongation at Break	Tensile Strength /MPa	Shore A Hardness /Shore A
Unfilled NR	2.23	598	14.6	41.6
${\mathrm{CB}}_{117}^{132}$	19.89	402	26.6	74.8
${\mathrm{CB}}_{145}^{105}$	14.58	530	27.7	70.8
${\mathrm{CB}}_{79}^{121}$	21.76	389	27.1	74.6
${\mathrm{CB}}_{111}^{108}$	16.69	491	27.3	71.5
${\mathrm{CB}}_{76}^{73}$	12.77	536	27.2	66.7
${\mathrm{CB}}_{96}^{55}$	8.33	620	28.0	63.7
${\mathrm{CB}}_{161}^{62}$	7.82	679	30.8	64.4
${\mathrm{CB}}_{37}^{84}$	15.38	461	23.1	66.1

**Table 4 polymers-14-01194-t004:** Results of multiple linear regression analysis of tensile properties–CB colloidal properties.

Regression Parameter	300% Modulus /MPa	Tensile Strength /MPa	Percent Elongation at Break	Shore A Hardness/Shore A
C	3.52	24.79	671.32	54.33
$\beta_{St}$	0.1657	−0.0232	−3.1103	0.1545
COAN p	3.47 × 10^−5^	0.1881	1.20 × 10^−6^	7.28 × 10^−5^
$\beta_{SA}$	−0.0404	0.0444	1.2572	0.0047
STSA p	0.0049	0.0091	1.91 × 10^−5^	0.6330
Adjusted R^2^	0.97	0.71	0.99	0.95

**Table 5 polymers-14-01194-t005:** Summary of CB selection recommendations to minimize mechanical hysteresis based on strain sweep and cyclic tensile tests.

Deformation Mode	Dynamic Deformation Conditions	Static Deformation Conditions
Strain control	Reduce CB surface area Reduce CB structure	Reduce CB structure
Stress control	Low Stress Levels Increase CB surface area Increase CB structure Medium-Large Stress Levels Reduce CB surface area Increase CB structure	Per Harwood, Mullins, and Payne, mechanical hysteresis at large strains under static stress-controlled conditions is independent of CB surface area and structure, although achieved elongations will depend on CB structure.
Strain energy control	Reduce CB surface area	Not determined in this study.

## Data Availability

The data presented in this study are available on request from the corresponding author.

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
