# Peer review of "The Influence of Colloidal Properties of Carbon Black on Static and Dynamic Mechanical Properties of Natural Rubber"

_polymers, 2022, doi:10.3390/polym14061194_

Round 1
Reviewer 1 Report
The article analysis the static and dynamic properties of natural rubber with carbon black. The article is written really well and is worth publishing in Polymers. Therefore, I have only one remark:
- There is no information about manufacturers or suppliers of raw materials used to prepare natural rubber without and with carbon black.
Author Response
Please see the attached letter to editor addressing the comments raised. Thank you very much.

Reviewer 2 Report
The work of Busfield and coworkers described the effect of the carbon black (CB) structure and surface area on the static and dynamic mechanical properties of the resulting rubber compounds. The research was an appropriate design, and the results were well presented. What I am missing here is a reason to be interested. For the presented study, electron microscopy, powder X-ray diffraction, and thermogravimetry should be applied to support the characteristic of the resulting rubber materials.
Comments:
- “A substantial body of historical experimental work has been performed in order to understand the role of CB in rubber reinforcement; particularly in regards to selection of CB and basic structure property correlations.”
It is not clear what this ‘basic structure property correlations’ means. The authors may consider the effect of CBs on the functional and material properties of the resulting rubber compounds.
- In the Introduction, the authors do not contain any information about the effect of CB crystallinity on the properties of rubber compounds. The crystallinity of the CBs used in work was also omitted.
- “For the purpose of this paper, a naming con-vention is adopted which allows the reader to immediately identify the type of CB based on its morphological properties and is given in Table 2.”
In my opinion, COAN described structural properties; therefore, the phrase ‘morphological properties’ is inadequate or should be modified. Morphological properties include the mean diameter of the particles, the surface area, and the mean diameter of the aggregate. In Table 2 the surface area was given.
Author Response

(The authors gave the same response as above.)
